Corrected: Publisher correction

# Copper adparticle enabled selective electrosynthesis of n-propanol

Jun Li [1,2], Fanglin Che[2], Yuanjie Pang[1,2], Chengqin Zou[2,3], Jane Y. Howe[4], Thomas Burdyny [1,5], Jonathan P. Edwards[1], Yuhang Wang [2], Fengwang Li [2], Ziyun Wang [2], Phil De Luna [6], Cao-Thang Dinh[2], Tao-Tao Zhuang[2], Makhsud I. Saidaminov[2], Shaobo Cheng[7], Tianpin Wu[8], Y. Zou Finfrock[8,9], Lu Ma [8], Shang-Hsien Hsieh [10,11], Yi-Sheng Liu [10], Gianluigi A. Botton [7], Way-Faung Pong[11], Xiwen Du[3], Jinghua Guo [10], Tsun-Kong Sham[12], Edward H. Sargent [2] & David Sinton [1]

The electrochemical reduction of carbon monoxide is a promising approach for the renewable production of carbon-based fuels and chemicals. Copper shows activity toward multi-carbon products from CO reduction, with reaction selectivity favoring two-carbon products; however, efficient conversion of CO to higher carbon products such as n-propanol, a liquid fuel, has yet to be achieved. We hypothesize that copper adparticles, possessing a high density of under-coordinated atoms, could serve as preferential sites for n-propanol formation. Density functional theory calculations suggest that copper adparticles increase CO binding energy and stabilize two-carbon intermediates, facilitating coupling between adsorbed *CO and two-carbon intermediates to form three-carbon products. We form adparticle-covered catalysts in-situ by mediating catalyst growth with strong CO chemisorption. The new catalysts exhibit an n-propanol Faradaic efficiency of 23% from CO reduction at an n-propanol partial current density of 11 mA cm$^{-2}$.

[1] Department of Mechanical and Industrial Engineering, University of Toronto, 5 King's College Road, Toronto, ON M5S 3G8, Canada. [2] Department of Electrical and Computer Engineering, University of Toronto, 10 King's College Road, Toronto, ON M5S 3G4, Canada. [3] Institute of New-Energy Materials, School of Materials Science and Engineering, Tianjin University, 300072 Tianjin, China. [4] Hitachi High Technologies America, Inc., 22610 Gateway Center Drive, Suite 100, Clarksburg, MD 20871, USA. [5] Materials for Energy Conversion and Storage, Department of Chemical Engineering, Delft University of Technology, 2629 HZ Delft, The Netherlands. [6] Department of Materials Science and Engineering, University of Toronto, 194 College Street, Toronto, ON M5S 3E4, Canada. [7] Canadian Center for Electron Microscopy, McMaster University, Hamilton, ON L8S 4M1, Canada. [8] Advanced Photon Source, Argonne National Laboratory, Lemont, IL 60439, USA. [9] Science Division, Canadian Light Source Inc., 44 Innovation Boulevard, Saskatoon, SK S7N 2V3, Canada. [10] Advanced Light Source, Lawrence Berkeley National Laboratory, Berkeley, CA 94720, USA. [11] Department of Physics, Tamkang University, 151 Yingzhuan Road, Tamsui District, 25137 New Taipei City, Taiwan, ROC. [12] Department of Chemistry, University of Western Ontario, 1151 Richmond Street, London, ON N6A 5B7, Canada. These authors contributed equally: Jun Li, Fanglin Che, Yuanjie Pang, Chengqin Zou. Correspondence and requests for materials should be addressed to E.H.S. (email: ted.sargent@utoronto.ca) or to D.S. (email: sinton@mie.utoronto.ca)

The electrochemical reduction of $CO_2$ to fuels using renewable electricity is of interest to close the carbon cycle[1–5]. Copper shows appreciable selectivity and activity for catalyzing the $CO_2$ reduction reaction ($CO_2$-RR) toward multi-carbon products[6–9]. During $CO_2$-RR, *CO (* denotes adsorbed species) is formed and is then further reduced and coupled to produce hydrocarbons and oxygenates[7,10–12]. The direct electrochemical reduction of CO (CO-RR) shares a similar product spectrum to $CO_2$-RR on Cu, and both approaches have seen significant progress in recent years[1,7,13,14].

In the creation of Cu electrocatalysts, strategies such as morphology control, facet tailoring, and oxygen engineering have been pursued to improve catalyst activity, selectivity, and durability[6,15–18]. Despite major advances in the efficiency and selectivity for $C_1$ and $C_2$ product electrosynthesis via $CO_2$/CO-RR, efficient carbon-chain upgrading to $C_3$ products has remained elusive.

N-propanol, a valuable $C_3$ alcohol that can be used as an engine fuel due to its high octane number of 118, has been reported in prior $CO_2$/CO-RR studies[13,15,19,20]. For example, an activated Cu mesh catalyst displayed a peak 13% n-propanol Faradaic efficiency (FE) from $CO_2$-RR at −0.9 V vs. a reversible hydrogen electrode (RHE)[20]. Prior work also reported 10% n-propanol FE using oxide-derived Cu catalysts from CO-RR at a much reduced applied potential of −0.4 V vs. RHE[13].

An challenge on the path toward more efficient and productive n-propanol production from $CO_2$/CO-RR arises from both the insufficient surface coverage of *CO and a large activation barrier for C–C coupling[21]. The inadequate stabilization of *$C_2$ intermediates on pristine Cu surfaces leads to desorption rather than further intermolecular reduction with *CO for $C_3$ generation. It is desirable to develop electrocatalysts that address these limitations.

Electrocatalytic conversion of CO is motivated by the potential to produce higher carbon products and the increasing availability of CO feedstock from industrial steel manufacturing and increasingly via $CO_2$ electroreduction to CO[3,22,23]. In particular, it has been shown that the two-step $CO_2$ to CO, then CO to $C_{2+}$ electrosynthesis outperforms the one-step $CO_2$ to $C_{2+}$ electroproduction based on established technoeconomic analysis owing to the more optimistic performance evaluation of each separate reaction and then overall better efficiency in the former case[24]. Additionally, feeding CO directly as the feedstock could boost the *CO adsorption on Cu and potentially facilitate the production of higher carbon products.

Here we report an adparticle structure that enables selective n-propanol electrosynthesis from CO-RR. Density functional theory (DFT) calculations suggest that the introduction of adparticles on metallic Cu surfaces facilitates $C_3$ formation due to the increase of *CO surface coverage and *$C_2$ stability, and the decrease of energy barriers towards intermolecular C–C coupling between *CO and *$C_2$. Experimentally, Cu adparticle electrocatalysts are achieved by in-situ electroreduction of an oxide precursor under a rich CO condition. This process, mediated by strong CO chemisorption, enables simultaneously rapid oxide reduction and adparticle growth as disclosed by operando X-ray absorption spectroscopy. The in-situ-derived Cu adparticle electrocatalysts exhibit the highest n-propanol FE of 23% ever achieved on Cu-based materials as well as an exceptional n-propanol partial current density of 11 mA cm$^{-2}$ at a moderate applied potential of −0.47 V vs. RHE.

## Results

### DFT calculations
In light of previous reports regarding the activation of metal surfaces by promoting surface cluster growth[25–28], we hypothesized that the introduction of adparticles, surface clusters possessing a high population of low-coordinated surface sites, on pristine Cu surfaces could drive $C_3$ production by increasing both the adsorption of CO and the binding energy of *$C_2$ intermediates. Specifically, increased surface concentrations of CO and *$C_2$ would decrease the reaction energy requirement for generating $C_3$ product (i.e., n-propanol). We employed DFT to assess the hypothesis that Cu adparticles could provide such benefits. We include a different number of adatoms on various Cu basal planes to construct one polycrystalline Cu adparticle model with a possibility of possessing various low-coordinated Cu sites (Fig. 1a; Supplementary Table 1).

We first examined CO adsorption energies on various Cu surfaces in the presence and absence of Cu adparticles (Supplementary Fig. 1). Our DFT calculations (Fig. 1b; Supplementary Figs. 2–4; Supplementary Table 2) clearly show that when CO adsorbs on the Cu adparticle, its adsorption can be significantly increased compared to adsorption over the pristine Cu surfaces ((i.e., Cu(100), Cu(111), and Cu(211)). Consequently, the modified Cu surfaces with Cu adparticles can increase CO adsorption densities compared to the pristine Cu surfaces.

As one of the most probable rate-limiting steps for generating *$C_2$ intermediates is the *CO dimerization[29,30], we then studied how Cu adparticle influences the reaction energies of the *CO dimerization reactions (Fig. 1c; Supplementary Figs. 5 and 6; Supplementary Table 3). The Cu adparticles on Cu(111) and Cu(211) surfaces show decreased reaction energies for the *CO dimerization step by ~0.8 and ~0.2 eV, respectively, compared to the bare (111) and (211) surfaces. In addition, for the most thermodynamically favorable flat Cu(111) surface, the Cu adparticle can stabilize the adsorption of *$C_2$ intermediates (i.e., *$CCH_2$ and *OCCOH) compared to the pristine Cu(111) surface (Supplementary Tables 4 and 5).

We further studied the effects of Cu adparticle on the reaction energy requirements of the possible intermolecular C–C coupling between *CO and *$C_2$ for n-propanol production. To date, few theoretical and experimental works have investigated reaction mechanisms for $CO_2$/CO-RR to n-propanol. From both theoretical and experimental point of views[30–32], the possible mechanisms of generating n-propanol can be narrowed down to the two likely coupling steps, CO–OCCOH and CO–$CCH_2$. Based on these previous findings, we computationally examined the CO–OCCOH and CO–$CCH_2$ coupling mechanisms over various Cu surfaces in the presence and absence of Cu adparticle via DFT calculations (Fig. 1d, e; Supplementary Figs. 7–10; Supplementary Tables 6 and 7). The results further indicate that the Cu adparticle on the most favorable Cu surfaces (i.e., Cu(111), Cu(100), and Cu(211) surfaces) could essentially decrease the reaction energies of C–C coupling of *CO and *$C_2$ intermediates proposed (i.e., *OCCOH and *$CCH_2$).

Taken together, the DFT calculations predict that Cu adparticle on pristine Cu surfaces exhibit enhanced catalytic effects (~0.4–0.8 eV decrease in reaction energy) for *CO dimerization or $C_1$–$C_2$ coupling. Some of the adparticle simulation results show a reduction from the base case, for example *CO dimerization on Cu(100) (Fig. 1c) or CO–OCCOH coupling on Cu(111) (Fig. 1d). The reduction, however, is ~0.1 eV, which is approximately the DFT self-interaction error. The simulations indicate that Cu adparticles decrease—on average—the reaction energy required to generating n-propanol during the CO-RR process. In addition, we varied the external fields and surface charge via the Neugebauer and Scheffler method[33], and the results support Cu adparticles enhancement of $C_3$ electroproduction during CO-RR (Supplementary Figs. 11–13).

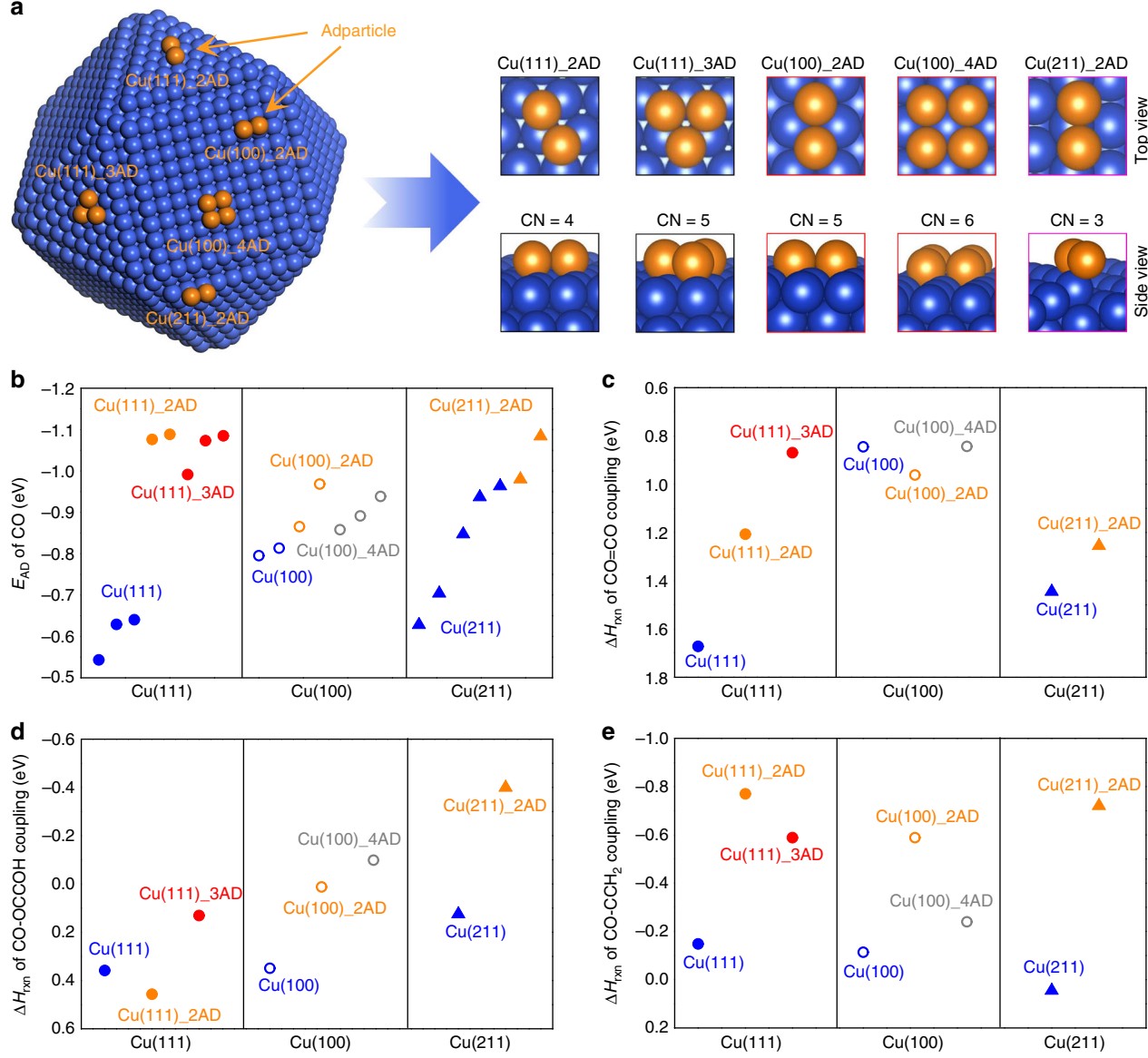

**Fig. 1** DFT calculations of Cu adparticle effects on CO-RR. **a** Demonstration of various low-coordinated Cu sites (CN: coordination number) by adding different number of Cu adatoms (ADs) on various Cu slabs. **b** The adsorption energy of CO. **c** The reaction energies of *CO dimerization. **d**, **e** CO-C$_2$ (i.e., *OCCOH, *CCH$_2$) on various modified Cu surfaces (i.e., (111), (100), (211)) in the presence and absence of Cu adparticles

**Synthesis and characterization of Cu adparticle catalysts.**
Inspired by DFT predictions, we sought to control experimentally the growth of Cu adparticles and investigate their impact on CO-RR selectivity and efficiency towards n-propanol production. CO interacts strongly with Cu surfaces and, in particular, chemically adsorbs on oxygen-covered Cu surfaces readily[26,27,34,35], resulting in oxygen removal from the Cu lattice and ultimately large-scale Cu surface reconstruction. Thus, we hypothesized that intense CO interaction with copper oxide during the initial electrochemical reduction with oxygen leaching and lattice restructuring would promote the simultaneous growth of adparticles from defects and under-coordinated sites. Accordingly, we prepared nanoparticulate copper oxide pre-catalysts (Fig. 2a–c) via a facile and scalable surfactant-directed wet chemistry method (see Methods for details)[36]. Then we directly deposited them onto a gas-diffusion electrode (GDE) embedded in a flow cell electrolyzer[7,37]. In contrast to an H-shaped electrocatalytic cell, the flow cell provides the high CO concentrations near the three-phase interface needed to synthesize

the adparticle-covered Cu catalyst (Fig. 2d), and ultimately facilitate CO conversion.

The in-situ-derived Cu catalysts share a similar overall size of ~300 nm with the oxide pre-catalysts (Fig. 2e). However, the surface roughness of the derived Cu catalysts increases significantly and an aggregated nanostructure morphology is observed, resulting from the intense surface reconstruction. High-resolution scanning electron microscopy shows the presence of densified adparticles as a form of aggregated nanoclusters (Fig. 2f, g). A number of possibly rich uncoordinated sites such as edge and corner atoms are also present[26], which, from our DFT studies, may function as surface active sites that catalyze CO-RR conversion to n-propanol formation (see below). Further evidence is offered by a high-resolution transmission electron microscopy examination (Fig. 2h–j), where adparticles, clearly distinguished from the Cu backbone, with an average size of ~3.2 nm are observed (Supplementary Fig. 14). Although the resulting adparticles are significantly larger than those simulated, particles of this size are expected to result in a variety of

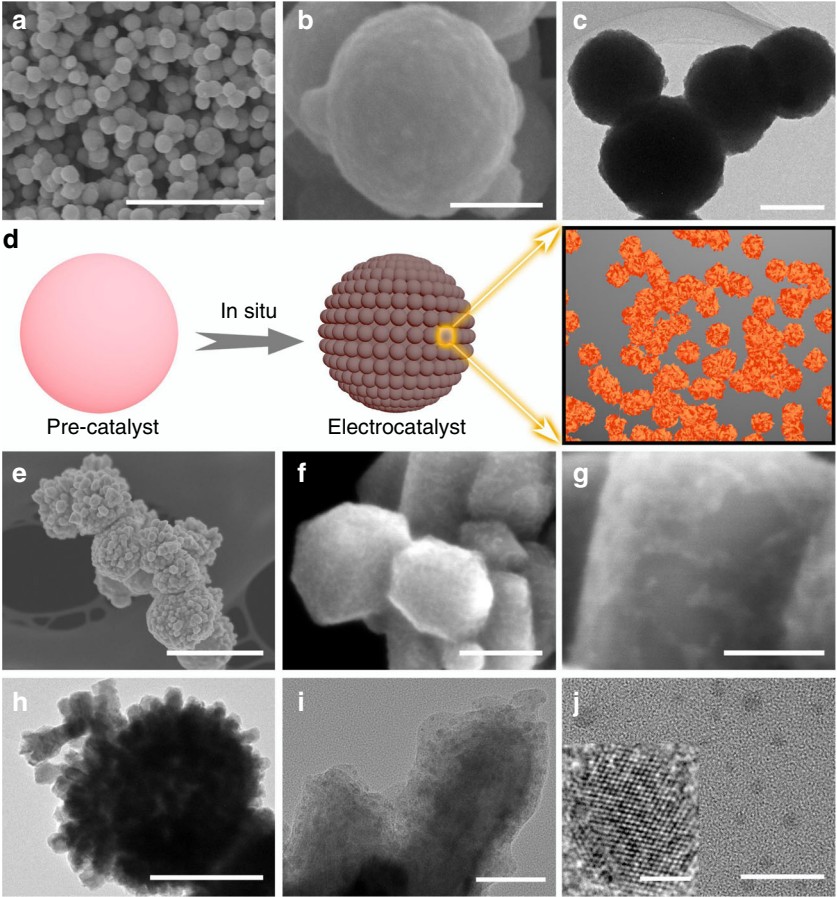

**Fig. 2** Structural characterization of pre-catalysts and Cu adparticles. **a–c** SEM/TEM images of oxide pre-catalysts. The scale bars are 2 μm in **a**, 200 nm in both **b** and **c**. **d** Schematic illustration of Cu adparticle growth from in-situ CO-RR. **e–g** SEM/HRSEM and **h–j** TEM/HRTEM images of Cu adparticle electrocatalysts obtained under CO-RR at −0.44 V vs. RHE in 1 M KOH. The scale bars are 500 nm in **e**, 200 nm in **h**, 50 nm in both **f** and **i**, 20 nm in both **g** and **j**, 2 nm in the inset of **j**

coordination numbers[38], as exemplified in the simplified, small-particle DFT simulations.

**Operando X-ray absorption spectroscopy**. To explore the in-situ growth of adparticles and their electronic nature, we employed operando hard X-ray absorption spectroscopy (hXAS) at the Cu K-edge. With this technique the real-time Cu valence states were tracked during the in-situ metallic reduction process. To verify the role of CO in the surface restructuring, we performed the experiments under CO and helium (inert) gas conditions using a flow cell setup identical to the one used for evaluating CO-RR performance (see below, Supplementary Fig. 15). For the case of Cu reduction under a CO gas environment, the oxide pre-catalysts, consisting dominantly of copper(I) oxide, are quickly reduced to a pure metallic Cu in less than 2 min at an applied potential of −0.44 V vs. RHE in a 1 M KOH electrolyte (Fig. 3a). Consistent results are illustrated by tracking the associated Cu bonding environment with the application of a Fourier transform filter (Fig. 3b). A linear combination fitting analysis of the Cu K-edge hXAS spectra was employed to quantitatively assess the Cu valence composition during the in-situ reaction. We found that the oxide-to-metallic Cu transition under CO gas is two-fold faster than that under He gas (Fig. 3c, d; Supplementary Fig. 16). This suggests that the intense CO chemisorption/reaction on oxygen-containing Cu surface during oxide reduction accelerates oxygen removal, promoting defect formation and driving Cu to reconstruct towards adparticle formation (Fig. 3e)[27]. Consequently, no trace of oxygen is resolved in fully-derived

Cu irrespective of applied gas condition, in agreement with ex-situ XRD results (Supplementary Fig. 17). A fitting analysis of the Cu hXAS spectra (Supplementary Fig. 18) further shows a lower Cu coordination in the derived Cu species compared to that in bulk Cu foil, indicating the presence of under-coordinated Cu sites in oxide-derived Cu. In addition, a further in-situ analysis using the more surface- and valence-sensitive Cu L₃-edge soft XAS (sXAS) (Supplementary Fig. 19)[6,39], compared to the Cu K-edge hXAS, also confirms the metallic Cu nature during CO-RR. These findings, taken together, demonstrate that the initial oxide reduction step can be boosted by CO chemisorption/ reaction on the oxygen-covered Cu surface, which then directs the growth of metallic adparticles.

**The CO-RR catalytic behavior**. Adparticle-enhanced n-propanol electrosynthesis is then tested by a CO-RR product analysis. Potential screening from −0.32 to −0.66 V vs. RHE of Cu adparticles (Fig. 4a) demonstrates a gradual enhancement of CO-RR over the hydrogen evolution reaction with more negative applied potentials. A sharp increase in the FE of $C_{2+}$ products ($FEC_{2+}$) from 35% to 72% is achieved within a narrow applied potential range from −0.32 to −0.44 V vs. RHE, and a maximum 89% of $FEC_{2+}$ is reached at −0.66 V vs. RHE. With increasing overpotential (Fig. 4b), the FE of the product of interest, n-propanol, first shows a dramatic increase from 2.8% at −0.32 V to 21% at −0.39 V vs. RHE, then reaches the peak value of 23% at both −0.44 V and −0.47 V vs. RHE, and finally drops to ~11% at −0.66 V vs. RHE. The decrease is likely due to the depletion

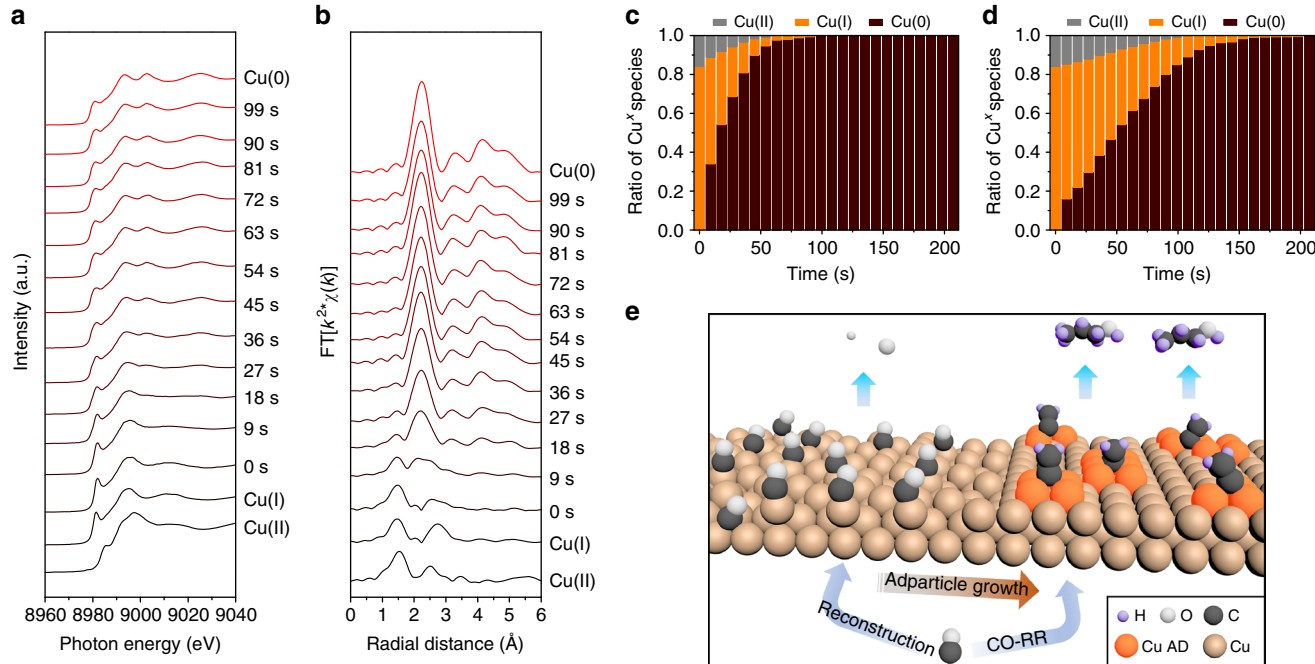

**Fig. 3** Operando Cu K-edge hXAS analysis. **a** Operando Cu K-edge hXAS for tracking the growth of Cu adparticles under CO-RR at −0.44 V vs. RHE in 1 M KOH. Cu K-edge hXAS of copper(II) oxide, copper(I) oxide and metallic Cu standards measured ex-situ are included for comparison. **b** Fourier transform of spectra in **a**. **c**, **d** Calculated ratio of Cu oxidation states over the course of oxide reduction at −0.44 V vs. RHE in 1 M KOH under CO and He gas condition, respectively, obtained by performing linear combination fitting of their corresponding Cu K-edge hXAS. **e** Schematic illustration of in-situ Cu adparticle (AD) growth mediated by CO chemisorption/reaction on defective Cu surface and the formation of n-propanol on adparticles

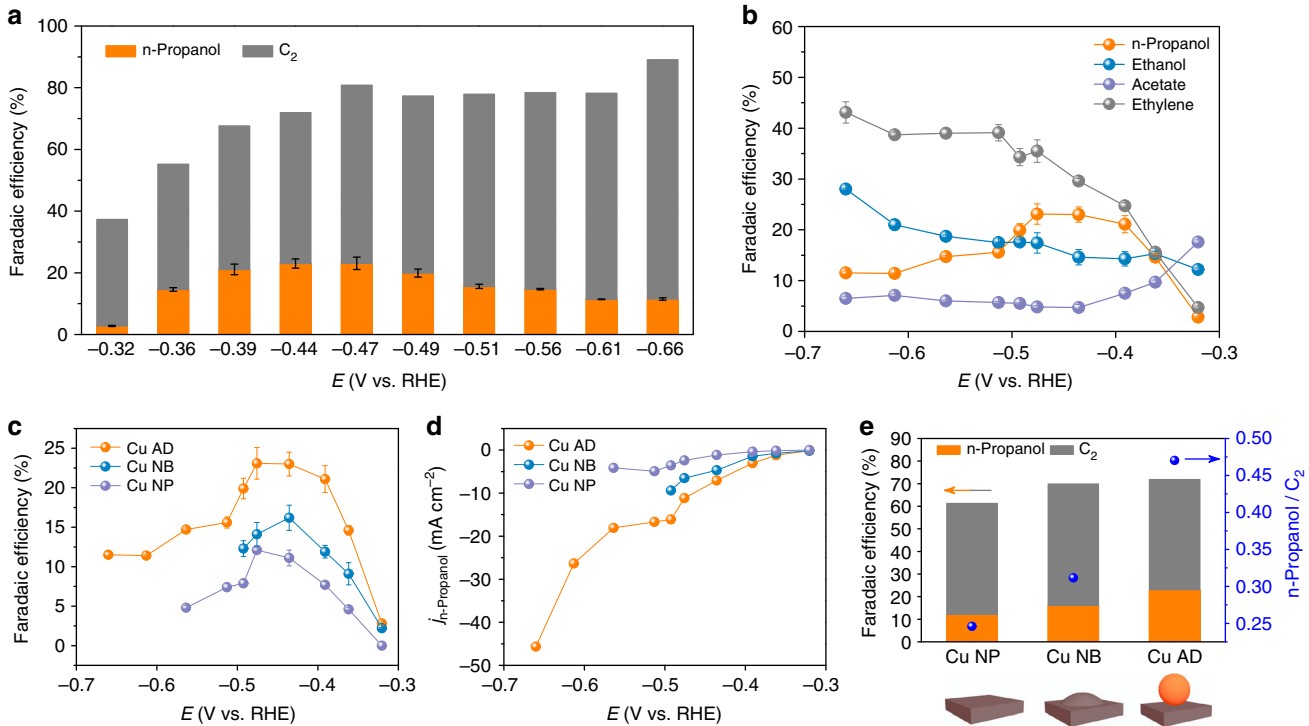

**Fig. 4** CO-RR performance on various Cu catalysts. **a**, **b** CO-RR product selectivities (FE, %) on Cu adparticle electrocatalysts at various applied potentials ($E$) in 1 M KOH. **c**, **d** CO-RR performance comparison in terms of n-propanol selectivities (FE, %) and activities (partial current density, mA cm$^{-2}$) among Cu adparticle (AD), Cu nanobump (NB), and Cu nanoparticle (NP) electrocatalysts at various applied potentials in 1 M KOH. **e** The peak FE$_{\text{n-propanol}}$/FE$_{C_2}$ ratio obtained on different electrocatalysts. Error bars are means ± SD ($n = 3$ replicates)

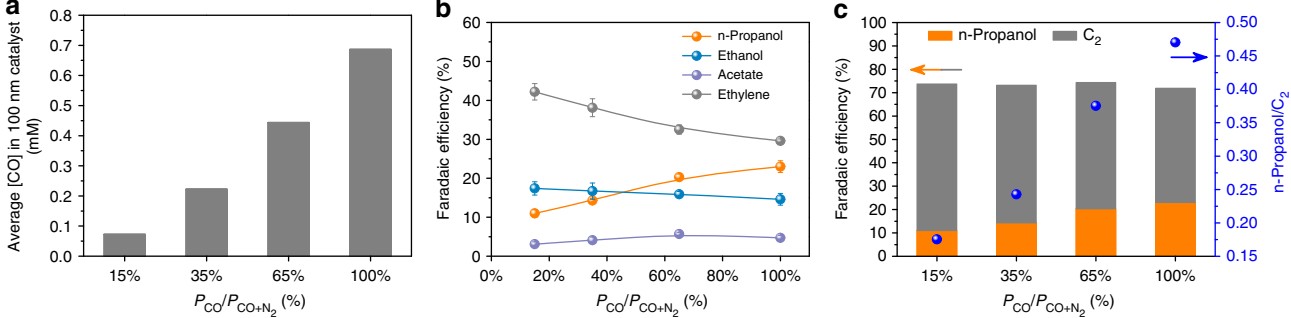

**Fig. 5** CO partial pressure analysis of n-propanol production. **a** Modeling of the average [CO] in 100 nm catalyst at 30 mA cm$^{-2}$ over the change of CO partial pressure in 1 M KOH; the similar current density was achieved experimentally with an applied potential of −0.44 V vs. RHE. **b** CO partial pressure-dependent test on Cu adparticle electrocatalysts at −0.44 V vs. RHE in 1 M KOH. **c** The FE$_{n\text{-propanol}}$/FEC$_2$ ratio of Cu adparticle electrocatalysts obtained on different CO partial pressure at −0.44 V vs. RHE in 1 M KOH

of CO at Cu surface at higher reaction rates (Supplementary Fig. 20). While both ethylene and ethanol exhibit an increase of FE with increasing overpotential, the FE of acetate first decreases and then reaches a plateau, which suggests a possible relationship between acetate and C$_{2+}$ chemicals. In particular, a prompt product switchover between acetate and ethylene/n-propanol at a narrow potential window from −0.3 to −0.4 V vs. RHE, indicates that acetate, ethylene, and n-propanol may share similar pathways. However, these pathways diverge depending on applied potentials, consistent with previous analysis[40].

To correlate the adparticle structure with the observed increase in n-propanol production from CO-RR, we compared performance against two different control samples. The first control was commercially available Cu nanoparticles (Cu NPs) with a size of ~100 nm. The formation of adparticles was not observed on these Cu NPs after CO-RR (Supplementary Fig. 21). The second control sample was prepared by in-situ reduction of oxide pre-catalysts under a nitrogen (inert) gas condition instead of using CO. In the absence of CO during the initial metallic reduction, the surface did not reconstruct to form adparticles, but instead formed rich grain boundaries on the surface of derived Cu (Supplementary Fig. 22). After CO-RR, the Cu surface displays an abundant amount of nanobumps (denoted as Cu NBs) with an average size of ~7.5 nm (Supplementary Fig. 23). These nanobumps appear to emerge upon the elimination of surface grain boundaries during the CO-RR reaction. The implication is that strong CO interactions during CO-RR with under-coordinated Cu sites can cause the samples that are reduced under purely N$_2$ conditions to restructure. This restructuring and elimination of grain boundaries relieves strain and forms undulate Cu NBs[26]; however, the formation of adparticles on Cu NBs is not observed (Supplementary Fig. 23). A direct comparison of the CO-RR performance among these three Cu species (Figs 4c, d; Supplementary Table 8) demonstrates the superior selectivity and activity of the Cu adparticle catalysts (Cu ADs; Supplementary Fig. 24) towards n-propanol production. Compared to the 23% FE for n-propanol achieved on the adparticle catalysts, the Cu NBs deliver a peak 16% of FE$_{n\text{-propanol}}$ at −0.44 V vs. RHE while Cu NPs show the least n-propanol selectivity with a maximum 12% of FE$_{n\text{-propanol}}$ obtained at −0.47 V vs. RHE. A further FE$_{n\text{-propanol}}$ over FEC$_2$ ratio comparison (Fig. 4e) suggests the selective n-propanol generation enabled by adparticles, where Cu AD presents a two-fold improvement over Cu NP. After normalizing to the electrochemical surface area, an intrinsic n-propanol activity of Cu AD catalysts is confirmed (Supplementary Figs. 25 and 26; Supplementary Table 9), in accordance with our DFT results showing

the decreased intermolecular coupling barriers of *CO and *C$_2$ intermediates on various Cu surfaces with the inclusion of adparticles. To further exclude additional morphology effects among Cu AD, NB, and NP species on CO-RR performance, we thermal-annealed Cu AD catalysts at 150 °C under N$_2$ gas condition. This processing erases the adparticle texture while maintaining the overall morphology (Supplementary Fig. 27). A dramatic decrease of FE$_{n\text{-propanol}}$ commensurate with the increase of FE$_{ethylene}$ is observed, further supporting the adparticle role in promoting n-propanol production. Compared to a nanobump (an undulate Cu surface), an adparticle (an isolated cluster) possesses a higher population of low-coordinated surface sites and thus a stronger CO chemisorption[38]. The larger surface area of an adparticle also has a higher chance of offering optimized geometries for CO adsorption and, therefore, more efficient n-propanol formation (Supplementary Fig. 2). By comparing to existing reports (Supplementary Table 10), adparticle-assisted n-propanol formation with an FE$_{n\text{-propanol}}$ up to 23% associated with a $j_{n\text{-propanol}}$ of 11 mA cm$^{-2}$ surpasses the best reported ~13% and 1.74 mA cm$^{-2}$, achieved separately, by a 1.8-fold selectivity and a six-fold activity enhancements. Additionally, an exceptionally high n-propanol partial current density of ~46 mA cm$^{-2}$ is also achieved from Cu AD at −0.66 V vs. RHE with an appreciable FE$_{n\text{-propanol}}$ of ~11%. These results, taken together, demonstrate that adparticle-covered Cu surfaces are active and selective for n-propanol generation from CO-RR.

To assess the n-propanol generation process, we tested the Cu AD catalysts at reduced CO partial pressures (Fig. 5). Note that the decrease of CO partial pressure in gas feedstock would directly reduce the local CO concentration at the Cu catalysts surface (Fig. 5a). Gas mixtures of CO and N$_2$ with a composition of 15%, 35%, and 65% CO, respectively, at 1 atm were introduced at a constant potential of −0.44 V vs. RHE in 1 M KOH. As CO partial pressure is increased, we observe an increase of FE$_{n\text{-propanol}}$ from 11% to 23% commensurate with a decrease of FE$_{ethylene}$ from ~42% to ~30% (Fig. 5b). The FE changes of ethanol and acetate are relatively small at all CO partial pressures. The ratio of FE$_{n\text{-propanol}}$ to FEC$_2$ (Fig. 5c) shows the increase of n-propanol production at the expense of C$_2$ (mainly ethylene). The shift from ethylene to n-propanol is consistent with *C$_2$ intermediates that contribute to ethylene formation (such as *OCCOH and *CCH$_2$; Fig. 1) being major intermediates for n-propanol as well. The similar ~11% FE$_{n\text{-propanol}}$ obtained on both Cu NP catalysts under 100% CO and Cu AD catalysts under 15% CO highlights the role of adparticles in concentrating CO and stabilizing *C$_2$ intermediates for n-propanol generation.

## Discussion

In this study, DFT and experimental results, taken together, portray an efficient Cu AD electrocatalyst enabling selective n-propanol production from CO-RR. It has been previously reported that derived Cu from the electrochemical reduction of oxide precursors would endure morphology reconstruction by exposing low-coordinated sites[6]. Meanwhile, strong CO interaction with low-coordinated Cu atoms would weaken the Cu–Cu bond and arouse large-scale Cu surface restructuring[25–27]. Therefore, the electroreduction of copper oxide with intense CO interaction enabled by the application of a flow cell could offer possibilities of forming new Cu structures, i.e., adparticle. Here we ascribed the adparticle growth to the CO-assisted rapid oxide reduction and Cu surface rearrangement under an in-situ flow cell CO-RR condition (Supplementary Fig. 15), in which abundant CO preferentially chemisorbs on defective sites during oxygen leaching, accelerating oxygen removal from the surface and driving Cu to reconstruct into adparticles as revealed from operando hXAS studies. Without the presence of intense CO reaction, the formation of dense grain boundary is observed (Supplementary Fig. 22a–c). It agrees with previous reports where oxide reduction carried out under a negligible CO environment (CO saturated KOH solution) results in grain boundary formation[13,18]. We further note that these grain boundaries are not stable under an intense CO reaction, which leads to the Cu surface restructuring towards nanobumps formation (Supplementary Fig. 22d, e) with the elimination of gain boundaries. Compared to Cu NP control, the inclusion of densified under-coordinated sites (e.g., adparticles and nanobumps) on Cu backbone significantly increases the density of active sites and accelerates the CO-RR reaction rates.

By carrying out a CO partial pressure analysis, we found a product formation trend where an increase of CO partial pressure favors n-propanol formation at the expense of $C_2$ (mainly ethylene) generation. It indicates that intermolecular C–C coupling of CO and ethylene precursors contributes to n-propanol formation, in agreement with our DFT results. An analysis between the CO partial pressure and $j_{n\text{-propanol}}$ shows their linear correlation with a slope of ~1 (Supplementary Fig. 28), suggesting the first-order dependence of $j_{n\text{-propanol}}$ on local CO concentration[41]. A proton concentration dependency on n-propanol production was also examined (Supplementary Fig. 29). With a range of 0.5–5 M KOH, 1 M KOH is found to produce the highest $FE_{n\text{-propanol}}$, a further increase of KOH concentration prioritizes the acetate formation due to the attack of n-propanol/ethylene intermediates in the concentrated $OH^-$ solution[13]. As previously reported, CO preferably bounds to low-coordinated Cu atoms[25,26] which also provide means of stabilizing intermediate species[18,42], insinuating the role of adparticle for concentrating CO and stabilizing $*C_2$ being further reduced to $C_3$. Indeed, by lowering the CO partial pressure to as low as 15%, Cu AD catalysts achieve an $FE_{n\text{-propanol}}$ of 11% (Fig. 5b) comparable to the $FE_{n\text{-propanol}}$ obtained on Cu NP controls under 100% CO condition (Fig. 4c). With this unique adparticle structure immobilized on bulky Cu surfaces, simulation results reveal that adparticle offers possibilities of increasing CO adsorption, promoting $*CO$ dimerization, stabilizing $*C_2$ intermediates, lowering energy barriers of intermolecular C–C coupling between $*CO$ and $*C_2$, and ultimately facilitating the formation of $C_3$ chemicals.

In summary, a control over under-coordinated sites is essential for optimizing hetero-electrocatalysis. We have demonstrated that adparticle-covered Cu catalysts capable of enhancing CO adsorption and stabilizing $*C_2$ intermediates. In CO-RR testing, Cu adparticle electrocatalysts promoted n-propanol production up to 23% $FE_{n\text{-propanol}}$ commensurate with an exceptional n-propanol partial current density of 11 mA cm$^{-2}$. This adparticle approach not only provides an avenue to high-chain carbon products formation under CO-RR, but may also be extended to improve the performance of other metal catalysts.

## Methods

**DFT calculations**. DFT calculations were performed using the Vienna Ab Initio Simulation Package (VASP) code[43,44]. To treat the exchange–correlation interactions and solve the ion–electron interactions in a periodic boundary system, we chose Perdew–Burke–Ernzerhof (PBE)[45] functionals and the projector-augmented wave (PAW) method[46]. We chose the most stable Cu(111) flat surface, the most active Cu(100) surface, and the most stable Cu(211) step surface in our study to validate the hypothesis that the Cu adparticles can decrease the reaction energies of the CO=CO dimerization and the $C_1$–$C_2$ coupling and, therefore, increase the Faraday efficiency (FE) of n-propanol production during electrochemical CO-RR. For the p($4 \times 4$) Cu(111) and Cu(100) supercell, a Monkhorst–Pack mesh[47] with a grid of ($3 \times 3 \times 1$) k points and a plane wave expansion up to 400 eV was employed. In addition, for the p($2 \times 4$) Cu(211) facet, a Monkhorst–Pack mesh[47] with a grid of ($4 \times 4 \times 1$) k points and a plane wave expansion up to 400 eV was used. All of the examined surfaces have a ~15 Å vacuum layer separation between each periodic unit cell. We also examined the CO adsorption energy (Supplementary Fig. 30) and reaction energy of CO dimerization (Supplementary Fig. 31) using RPBE functional, which is more accurate for the catalytic intermediate adsorption on single-crystal late transition-metal surfaces that involve pure chemisorption (i.e., CO over Cu)[48,49]. The main trend that Cu adparticles increase the adsorption strength of CO and decrease the reaction energies of CO dimerization is supported by the RPBE functional results. Computational details are included in the Supplementary Methods.

**Preparation of oxide pre-catalysts**. All reagents used in this work were from Sigma Aldrich without further purification. A surfactant-directed synthesis method was used to prepare the copper oxide precursor[36]. In general, 4.2 g triblock copolymer Pluronic P123 (MW 5800) was uniformly dispersed in 210 mL deionized water under a continuous stirring in a water bath at room temperature (~20 °C) to make clear solution I. Separately, 0.4 g copper(II) fluoride ($CuF_2 \cdot 2H_2O$) was dissolved in 20 mL aqueous solution with the inclusion of 3 mL concentrated ammonia solution (~30%) under constant stirring to prepare dark blue solution II. Then solution II was quickly added to solution I under stirring for 30 min to obtain dark blue solution III. During this time, a clear solution IV containing 0.6 M L-ascorbic acid was made. Later, ~45 mL solution IV was added into solution III in a dropwise manner to obtain a final orange suspension. The final product was collected by repeating three times of centrifuging and ethanol rinsing, and dried overnight under vacuum at room temperature (~20 °C).

**Preparation of GDE**. Ten milligrams of as-made oxide pre-catalysts were firstly dispersed in 1 mL methanol with the inclusion of 40 μL Nafion solution (~5 wt%) under constant stirring for 15 min, As-prepared suspension was then air-brushed onto a $2 \times 4$ cm$^2$ (cut into two pieces after two separate tests) Freudenberg gas-diffusion layer (GDL; Fuel Cell Store) using nitrogen as the carrier gas to make GDE, which was vacuum dried for at least 5 h before use. The loading amount of oxide precursor was determined to be ~0.75 mg cm$^{-2}$ by the net weight gain before and after air-brushing.

**Characterization**. Surface morphology was analyzed using a Hitachi SU9000 SEM/STEM at 2 kV. A Hitachi HF-3300 instrument with an acceleration voltage of 100 kV was employed for TEM analysis. For ultrahigh resolution TEM work an aberration-corrected FEI Titan 80-300 was used (200kV). XRD was performed on a MiniFlex600 instrument with a copper target ($\lambda = 1.54056$ Å) at room temperature. Operando hXAS measurements were conducted at 9BM and 20BM beamlines of the Advanced Photon Source (APS, Argonne National Laboratory, Lemont, Illinois), in partnership with the Canadian Light Source (CLS) in Saskatoon (Saskatchewan). In-situ soft X-ray absorption spectroscopy was carried out at the 8.0.1 beamline of the Advanced Light Source (ALS; Lawrence Berkeley National Laboratory, Berkeley, California).

**Electroreduction of carbon monoxide**. All CO reduction experiments were carried out in a gas–catholyte–anolyte three-compartment flow cell architecture[7] using a three-electrode system, where as-made GDE, Ag/AgCl (with 1 M KCl solution filled), and nickel foam (1.6 mm thickness; MTI Corporation) were used as cathode, reference electrode, and anode, respectively. The catholyte and anolyte compartments were separated by an anion exchange membrane (Fumasep FAA-PK-130). The cell was powered by an electrochemical workstation (Autolab PGSTAT302N), and an electrolyte of 1 M KOH was used all times. During the CO-RR test, 1 M KOH was directed separately into the cathode and anode compartments using two variable-speed peristaltic tubing pumps (Control Company 3385) and circulated through the electrochemical cell. CO gas was flowed through the gas compartment at the backside of GDE with a flow rate of 30 standard cubic centimeters per minute (s.c.c.m.). Applied cathode potentials after iR compensation were converted to the RHE reference scale using $E_{RHE} = E_{Ag/AgCl} + 0.235$ V +

$0.059 \times \text{pH}$. In this study, both liquid and gas products were collected for identifications and quantifications. Liquid product dissolved in catholyte was identified and quantified with one-dimensional $^1$H nuclear magnetic resonance spectroscopy ($^1$H NMR) using a Agilent DD2 500 spectrometer, in which a diluted dimethyl sulfoxide in $D_2O$ was used as an internal standard (Supplementary Fig. 32). Gas outlet from flow cell experiment was analyzed with a gas chromatograph (GC; PerkinElmer Clarus 680) equipped with a Molecular Sieve 5A capillary column and a packed Carboxen-1000 column. By using Argon (Linde, 99.999%) as a carrier gas, a thermal conductivity detector and a flame ionization detector equipped with a methanizer loaded in GC instrument were used to quantify hydrogen and ethylene, respectively.

**Modeling of CO diffusion across the gas-diffusion layer**. The concentration of CO in the electrolyte and present at the catalyst's surface was determined using a 1D reaction-diffusion model. The model is adapted from a previous report which modeled the diffusion of $CO_2$ across a gas-diffusion layer for various electrolytes, current densities, and pressures[7]. Unlike the $CO_2$ model, CO diffusion into the electrolyte at 0 mA cm$^{-2}$ does not directly affect the local pH or change the concentration of CO or OH$^-$ as a function of distance into the electrolyte. The governing equations for the simulation are shown below and extend from the gas–liquid interface of the gas-diffusion layer at $x = 0\,\mu\text{m}$ to an assumed diffusion layer boundary thickness of $x = 500\,\mu\text{m}$ into the electrolyte.

$$\frac{\partial [CO]}{\partial t} = D_{CO}\frac{\partial^2 [CO]}{\partial x^2} - R_{CO} \tag{1}$$

$$\frac{\partial [OH^-]}{\partial t} = D_{OH^-}\frac{\partial^2 [OH^-]}{\partial x^2} + R_{OH} \tag{2}$$

where $R_{CO}$ and $R_{OH}$ account for the consumption of CO in the reduction reaction and the production of OH$^-$, respectively. These reactions are assumed to occur homogeneously throughout the catalyst layer such that the source and sink of CO and OH$^-$ are spatially dependent:

$$R_{CO} = \begin{cases} \frac{j}{F}\left(\frac{FE_{\text{ethylene+ethanol}}}{n_{\text{ethylene}}} + \frac{FE_{\text{acetate}}}{n_{\text{acetate}}} + \frac{FE_{\text{n-propanol}}}{n_{\text{n-propanol}}}\right)\frac{\varepsilon}{L_{\text{catalyst}}}, & 0 \leq x \leq L_{\text{catalyst}} \\ 0, & x > L_{\text{catalyst}} \end{cases} \tag{3}$$

$$R_{OH} = \begin{cases} \frac{j}{F}\frac{\varepsilon}{L_{\text{catalyst}}}, & 0 \leq x \leq L_{\text{catalyst}} \\ 0, & x > L_{\text{catalyst}} \end{cases} \tag{4}$$

where $F$ is the Faraday's constant and is taken as 96485 C mol$^{-1}$ and $j$ is the geometric current density. As was done previously, a catalyst layer porosity, $\varepsilon$, of 60% was assumed. Based upon the experimental results, product selectivities of 15% hydrogen, 50% ethylene+ethanol, 10% acetate, and 25% n-propanol are assumed for all simulations to approximately account for the number of electrons transferred per CO molecule consumed. A reaction thickness of 100 nm was assumed for the catalyst layer ($L_{\text{catalyst}} = 100$ nm). The maximum solubility of CO in the electrolyte was modeled by Henry's constant at 1 atm and 298 K as well as taking salting out effects into account via the Sechenov equation.

A no-flux boundary condition was imposed at the left-hand boundary for OH$^-$ while the concentration of CO was initially described by the solubility of CO in the imposed KOH concentration and partial pressure conditions. Both CO and KOH were fixed to their bulk electrolyte concentrations at the boundary layer thickness of 500 μm. The concentration profiles of CO and OH$^-$ were then found at steady state for a variety of KOH concentrations, partial pressures, and current densities.

## Data availability
The data that support the findings of this study are available from the corresponding author on reasonable request

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

## Acknowledgements

This work was financially supported by the Ontario Research Fund Research-Excellence Program, the Natural Sciences and Engineering Research Council (NSERC) of Canada, the CIFAR Bio-Inspired Solar Energy program, and the University of Toronto Connaught grant. Synchrotron work was carried out at the Advanced Photon Source (APS) in partnership with the Canadian Light Source (CLS), and the Advanced Light Source. APS in Argonne National Laboratory and the ALS in Lawrence Berkeley National Laboratory are supported by the Office of Basic Energy Sciences of the US Department of Energy under Contract No. DE-AC02-06CH11357 and No. DE-AC02-05CH11231, respectively. CLS is funded by NSERC, the Canada Foundation for Innovation (CFI), National Research Council (NRC), Canadian Institute for Health Research (CIHR), and the University of Saskatchewan. Part of the HRTEM work (Titan 80-300) was carried out at the Canadian Centre for Electron Microscopy, a facility supported by CFI under the Major Science Initiative program, NSERC and McMaster University. All DFT computations were performed on the IBM BlueGene/Q supercomputer with support from the Southern Ontario Smart Computing Innovation Platform (SOSCIP). SOSCIP is funded by the Federal Economic Development Agency of Southern Ontario, the Province of Ontario, IBM Canada Ltd., Ontario Centres of Excellence, Mitacs and 15 Ontario academic member institutions. The authors thank Dr. M.J. Ward, Dr. Z.Q. Wang, and Mr. J.T. Cheng for technical support at 20BM beamline of APS. C.Z. gratefully acknowledges a scholarship under the International Academic Exchange Fund for Joint Ph.D. Student from Tianjin University. D.S. gratefully acknowledges support from a NSERC E.W.R. Steacie Memorial Fellowship. T.B. thanks Hatch for a Graduate Scholarship for Sustainable Energy Research. P.D.L. thanks NSERC for financial support in the form of the Canada Graduate Scholarship—Doctoral (CGS-D) award. J.L. and M.I.S. acknowledge the Banting Postdoctoral Fellowships program.

## Author contributions

E.H.S. and D.S. supervised the project. J.L., F.C., Y.P. and C.Z. designed and carried out all the experiments. F.C. performed and Z.W. helped the DFT simulation. C.Z. and X.D. collected the SEM, TEM, and HRTEM images. J.Y.H. conducted the HRSEM characterization. S.C. and G.A.B. conducted some of the HRTEM analyses. T.B. simulated the diffusion-reaction. M.I.S. performed the XRD test. J.L. carried out all the XAS measurements and data analysis with the assistance of T.W., Y.Z.F., L.M., S.-H.H., W.-F.P., Y.-S.L. and J.G., and T.-K.S. J.P.E., Y.W., F.L., P.D.L., C.-T.D. and T.-T.Z. helped data analysis and manuscript polishing. All authors discussed the results and assisted during manuscript preparation.

## Additional information

**Competing interests:** The authors declare no competing interests

