## [Peer Review File · Nature Communications]

Reviewers' comments:

Reviewer #1 (Remarks to the Author):

The manuscript by Li and co-workers describes Cu adparticles covered on a gas diffusion electrode that produces n-propanol in electrochemical CO conversion. The importance of high energy surface at under-coordinated atoms has been numerously emphasized in the catalyst research society, however, its application in CO conversion reaction and high product selectivity for value-added liquid fuel (n-propanol, ~23%) are unprecedented to this reviewer's knowledge. The synthetic strategy of using strong CO chemisorption environment and growing the adparticles at in-situ condition, can be applied to various catalysts and energy devices. However, in spite of such phenomenological interest, the understanding of reaction mechanism is deficient and yet supported fully by the evidence of the analysis. Additional experiments and explanation of the resulted phenomenon seems required before its publication. The suggestions are as follows.

Major points

- 1) Most of all, this reviewer would like to indicate that the correlation between DFT result and experimental observation is yet to be fully explained. By using DFT calculations, the author claimed that the modified Cu surfaces with Cu adparticle can stabilize *CO and *C2 intermediates simultaneously, compared with Cu surface and thereby, enhance the C3 chemical formation. Especially, the detailed effects of 2, 3 and 4 adparticle on Cu (111), (110), (211) plane is described. However, in the synthesis and characterization section, only the presence of densified adparticles was suggested by HR-SEM and HR-TEM, without any further evidence of their detailed structure and catalytic effect.
- 2) Especially, according to the Ex-situ XRD spectra in Supplementary Fig. 14, (111) and (200) are the major plane in the synthesized electrode. Based on the DFT calculation, those planes can exhibit both enhanced and reduced catalytic effect, depending on the size of adparticle (Figure 1. c, d, e).
- 3) In addition, some of the DFT results are ambiguous. For example, the author explained that Cu adparticle can have higher *CO surface density, as it exhibits positively higher value for *CO adsorption energy (Supplementary Table 2). At the same time, the author also claimed that the Cu adparticle can stabilize the *C2 intermediates, although it has negatively higher adsorption energy (Supplementary Table 4 and 5).
- 4) One of the major claims in this paper is intense interaction between CO and Cu oxide promotes the simultaneous growth of adparticles, which accelerates CO to C2+ chemical conversion. For its verification, the author compared catalytic performance of Cu nanobump and Cu adparticles, prepared in N₂ (inert) and CO environment, respectively, and explained the phenomena based on the DFT calculation again. In addition to the computational simulation, I hope the authors validate their claim by experimental evidence also. For instance, temperature-programmed desorption (TPD) can be an approach (J. Am. Chem. Soc., 2015, 137, 9808).
- 5) By controlling the CO partial pressure, the author suggested that intermolecular C-C coupling of CO and ethylene precursors mainly contributes to n-propanol formation. I think more discussions can be deduced from this result. For instance, by plotting the partial current for n-propanol formation against the local CO concentration, reaction order for [CO] can be suggested (ACS Catalysis. 2018, 8, 931).
- 6) In the same context, proton concentration dependency on n-propanol production also can be a useful key to elucidate the reaction mechanism. While both of *OCCOH and *CCH₂ are currently

suggested as the possible source of C2 precursor, those intermediates necessitates different number of protons for its formation. As the author indicated in the manuscript, since the CO concentration is mainly determined by CO partial pressure in the gas feedstock, this can be done by simply changing the pH of electrolyte.

7) I also would like to point out the possible relationship between acetate and C2 chemicals, and/or acetate and n-propanol. Although the Faradaic efficiency change of acetate is relatively small at all CO partial pressures (Figure 5), however, all of ethylene, ethanol and n-propanol exhibit a general increase of Faradaic efficiency with increasing overpotential, while that of acetate decreases against applied potential (Figure 4b). This may imply subsequent conversion of CO to acetate, then to other C2 and C3 chemicals.

Reviewer #2 (Remarks to the Author):

In this paper, the authors put forward a Cu “adparticle” structure, both computationally and experimentally, as being particularly selective for CO electrocatalytic reduction to n-propanol. They start with a DFT hypothesis that clusters of 2-4 Cu atoms sitting on top of a single crystal Cu surface can potentially serve as active sites for CO reduction and utilize this structure to model the adsorption energies of key intermediates. They then synthesize a Cu “adparticle” catalyst that consists of ~3-5 nm Cu nanoparticles sitting on larger Cu particles and study its CO reduction activity. They compare this system to two other Cu nanostructured catalysts and observe an increase in peak n-PrOH faradaic efficiency from 15% to 23% on Cu AD vs. Cu NB at -0.54 V vs. RHE.

The 23% faradaic efficiency for n-PrOH and partial current density of 11 mA/cm² represents an improvement in absolute selectivity and activity compared to literature examples for this reaction. However, the structural hypothesis that they put forward as being responsible for this change in activity is not strongly substantiated. For modest activity differences like the ones reported here, it is difficult to clearly attribute that difference to a specific structure without a wide range of comparison materials.

A few specific comments are provided below:

1. The DFT calculations put forward a very specific structure, i.e. 2, 3, or 4 Cu atoms sitting on top of a single-crystal Cu surface. While the calculations provide interesting data about adsorption energies, the connection to the experimental samples are quite tenuous. The actual “adparticle” structure studied consists of 3-5 nm nanoparticles on a Cu surface, which represents hundreds of Cu atoms. How does the experimental structure actually relate to the DFT structural model? Is it possible to extrapolate conclusions from a 2-4 adatom structure to an adsorbed nanoparticle with hundreds of atoms?

2. The authors mention that their catalysts have a higher ECSA than oxide-derived Cu samples produced in other ways. To what extent is the ECSA affecting the surface-area normalized current density? Several authors have shown that mass-transport within a high-surface area material can affect the selectivities for H₂ vs. C₁ products vs. C₂ products. Is there also a potential correlation between ECSA and C₂ vs. C₃ products? This may also be consistent with the observed increase in C₃ selectivity with higher CO partial pressures.

3. The authors hypothesize that increased CO binding at undercoordinated Cu surface sites may be responsible for increased C₃ selectivity on their Cu AD samples. Besides DFT, have the authors been

able to measure a CO adsorption energy on their surface?

4. The authors compare their samples to a Cu “nanobump” sample and a Cu nanoparticle sample. Since all three of these materials are prepared in different ways and result in very different morphologies, it is difficult to build a structure-activity relationship just from these three materials. Is there any way to systematically vary the ad-nanoparticle size or density using their synthetic method to obtain more systematic structure-activity relationships?

In summary, the authors have put forward a nanostructured Cu catalyst with increased n-PrOH selectivity compared to other nanostructured Cu catalysts. The catalytic data for the increased selectivity is clear, thorough, and compelling. However, the connection between the structure and catalytic activity based on DFT and two comparison samples is much less clear. As a result, I believe this paper would be more suitable for a catalysis specific journal.

Reviewer #3 (Remarks to the Author):

Li et al. present a strategy to enhance the production of 3-carbon products in electrochemical CO reduction on copper by introducing under-coordinated surface copper atoms that enhance the binding of CO and 2-carbon intermediates in close proximity. They motivate this idea using density-functional theory calculations of adsorption and reaction energies, synthesize particles with suitable under-coordinated surface atoms and demonstrate selectivity to 3-carbon products in electrochemical reduction. This manuscript makes an important contribution towards favoring longer-chain products in electro-catalysis, a key scientific challenge, and merits publication in Nature Communications.

Some questions the authors should address prior to publication:

1. Electrochemical environments can strongly affect adsorption and reaction energies, especially by favoring charged surface configurations. This is now increasingly being incorporated in DFT calculations using advanced solvation models and grand-canonical (fixed electron potential) techniques to predict accurate onset potentials (eg. in Refs 9, 28 cited in the manuscript). The authors should justify their choice / acknowledge the limitation of neutral in-vacuo DFT calculations.
2. CO adsorbed on metal surfaces is a prototypical example of where DFT can be rather inaccurate, completely mis-predicting the stable adsorption sites and geometries. The authors should acknowledge this limitation as well.
3. The term adparticle seems suitable for the on-average 3 nm sized entities observed in the high-resolution TEM images, while the calculations seem to be on small clusters of adatoms which are less than a nm across. Given this discrepancy in scale, would edges on stepped surfaces be more representative of the active sites in the experimentally-observed structures, rather than islands of few atoms?

Thank you for a constructive review process. We have revised the work as detailed below:

Reviewer #1 (Remarks to the Author):

The manuscript by Li and co-workers describes Cu adparticles covered on a gas diffusion electrode that produces n-propanol in electrochemical CO conversion. The importance of high energy surface at under-coordinated atoms has been numerously emphasized in the catalyst research society, however, its application in CO conversion reaction and high product selectivity for value-added liquid fuel (n-propanol, ~23%) are unprecedented to this reviewer's knowledge. The synthetic strategy of using strong CO chemisorption environment and growing the adparticles at in-situ condition, can be applied to various catalysts and energy devices. However, in spite of such phenomenological interest, the understanding of reaction mechanism is deficient and yet supported fully by the evidence of the analysis. Additional experiments and explanation of the resulted phenomenon seems required before its publication. The suggestions are as follows.

Author's Reply: We have acted on each concern, including the requirement that we improve the linkages between DFT and experiment results. We have also pursued further experimentation and analysis to challenge our hypothesis. We detail our actions below.

Major points

1) Most of all, this reviewer would like to indicate that the correlation between DFT result and experimental observation is yet to be fully explained. By using DFT calculations, the author claimed that the modified Cu surfaces with Cu adparticle can stabilize *CO and *C₂ intermediates simultaneously, compared with Cu surface and thereby, enhance the C₃ chemical formation. Especially, the detailed effects of 2, 3 and 4 adparticle on Cu (111), (110), (211) plane is described. However, in the synthesis and characterization section, only the presence of densified adparticles was suggested by HR-SEM and HR-TEM, without any further evidence of their detailed structure and catalytic effect.

Author's Reply: We have now clarified the manuscript on this point. The inclusion of a different number of adatoms on various Cu basal planes in the simulations was not to address the size effect of adparticles in different cases. Rather, we build a one polycrystalline Cu adparticle model with a possibility of possessing various low-coordinated Cu sites (**Supplementary Table 1**), to demonstrate the low-coordination effect in general, and the associated enhancement in the C₃ chemical formation, consistent with our experimental results.

We have made these changes to the manuscript accordingly:

- (a) **Fig. 1a** has been revised to clearly show the geometry here and, most importantly, the adparticles of interest and their associated coordination numbers. This was not sufficiently clear previously, and the original **Fig. 1** simulation schematics gave the impression that the adparticle size was the key parameter probed.
- (b) We are also now clear in the text. **Page 6, Line 4**: “We employed density functional theory (DFT) to assess our hypothesis that Cu adparticles could provide such benefits. We include a different number of adatoms on various Cu basal planes to construct one polycrystalline Cu adparticle model with a possibility of possessing various low-coordinated Cu sites (**Fig. 1a, Supplementary Table 1**).”

2) Especially, according to the Ex-situ XRD spectra in Supplementary Fig. 14, (111) and (200) are the major plane in the synthesized electrode. Based on the DFT calculation, those planes can exhibit both enhanced and reduced catalytic effect, depending on the size of adparticle (Figure 1. c, d, e).

Author’s Reply: The DFT results show that the main (111) and (100) planes exhibit overall large enhanced catalytic effects (~0.4 eV - 0.8 eV decrease in reaction energy) for CO dimerization and C₁-C₂ couplings (**Fig. 1c, d, e**). We reason that the slightly reduced catalytic effects (~0.1 eV increase in reaction energy) for CO dimerization on Cu(100)_2AD (**Fig. 1c**) or C₁-C₂ coupling on Cu(111)_2AD (**Fig. 1d**) are within the typical DFT self-interaction error. Overall, the DFT results indicate that Cu adparticles can thermodynamically enhance the productivity of C₃ products during CO-RR.

We now clarify in this in manuscript as below. **Page 7, Line 11**:

“Taken together, the DFT calculations predict that Cu adparticle on pristine Cu surfaces exhibit enhanced catalytic effects (~0.4 – 0.8eV decrease in reaction energy) for *CO dimerization or C₁-C₂ coupling. Some of the adparticle simulation results show a reduction from the base case, for example *CO dimerization on Cu(100) (Fig. 1c) or CO-OCCOH coupling on Cu(111) (Fig. 1d). The reduction, however, is ~0.1 eV which is approximately the DFT self-interaction error. The simulations indicate that Cu adparticles decrease – on average – the reaction energy required to generating n-propanol during the CO-RR process.”

3) In addition, some of the DFT results are ambiguous. For example, the author explained that Cu adparticle can have higher *CO surface density, as it exhibits positively higher value for *CO adsorption energy (Supplementary Table 2). At the same time, the author also claimed that the Cu adparticle can stabilize the *C₂ intermediates, although it has negatively higher adsorption energy (Supplementary Table 4 and 5).

Author's Reply: In light of reviewer's feedback, and to rectify the inconsistency, we now make the definition of the adsorption energies for *CO and *C₂ intermediates clear and consistent. It is now clear that both energies are negative, indicating that both *CO and *C₂ adsorption energies are negatively and of greater magnitude in the adparticle case.

In supplementary information (SI), we modified **Equation S1** of the revised manuscript ' $E_{ad} = E_{(CO/slab)} - E_{(slab)} - E_{(CO(gas))}$ '. We then modified the description of this equation "Here, the more negative value of the adsorption energy represents a stronger binding strength of the CO molecule."

We also replotted **Fig. 1b** and **Supplementary Table 2** in the revised manuscript and SI with new definition of CO adsorption energies.

4) One of the major claims in this paper is intense interaction between CO and Cu oxide promotes the simultaneous growth of adparticles, which accelerates CO to C₂+ chemical conversion. For its verification, the author compared catalytic performance of Cu nanobump and Cu adparticles, prepared in N₂ (inert) and CO environment, respectively, and explained the phenomena based on the DFT calculation again. In addition to the computational simulation, I hope the authors validate their claim by experimental evidence also. For instance, temperature-programmed desorption (TPD) can be an approach (J. Am. Chem. Soc., 2015, 137, 9808).

Author's Reply: First, we returned to the lab to prepare enough derived-Cu powder samples for CO-TPD measurements (~100 mg/sample/test). The results of the CO-TPD testing are now shown in **Figure R1-1**. Both Cu AD and NB samples exhibited strong desorption signals at high temperature compared to Cu NP controls; however, it was not possible to confirm that this desorption was CO, and not contamination. Although the samples were stored under inert N₂ gas condition before testing, contamination from carbon species from the gas diffusion layer (porous carbon materials) or partially oxidized copper cannot be ruled out when using this complex integrated sample. We received similar reply regarding the CO-TPD test for this material system from Prof. Ib Chorkendorff at DTU who contributed to the JACS work mentioned by the reviewer.

Second, in order to highlight the significance of adparticle for CO adsorption and hence C₃ formation under CORR experimentally, we conducted a post-annealing process on our Cu AD sample to erase the adparticle texture while maintaining the overall morphology of Cu catalysts (**Figure R1-2**). A shift of product selectivity from n-propanol to ethylene after annealing is observed compared to Cu AD results in **Fig. 4**, which is due to the thermal removal of the AD texture.

Figure R1-1. CO-TPD profiles of Cu AD, NB and NP samples.

Figure R1-2. SEM images (a, b) of Cu AD post-annealed at 150 °C for 2h under N_2 and its associated CO-RR product distributions (c, d).

5) By controlling the CO partial pressure, the author suggested that intermolecular C-C coupling of CO and ethylene precursors mainly contributes to n-propanol formation. I think more discussions can be deduced from this result. For instance, by plotting the partial current for n-propanol formation against the local CO concentration, reaction order for [CO] can be suggested (ACS Catalysis. 2018, 8, 931).

Author's Reply: We found that, in the CO partial pressure range from 15% to 100%, a linear correlation exists between $j_{n\text{-propanol}}$ and CO partial pressure, with a slope of

~1 (**Figure R1-3**), suggesting a first-order dependence of $j_{n\text{-propanol}}$ on local CO partial pressure.

We also added this discussion into the revised manuscript. **Page 14, Line 10:**

“An analysis between the CO partial pressure and $j_{n\text{-propanol}}$ shows their linear correlation with a slope of ~1 (**Supplementary Fig. 28**), suggesting the first-order dependence of $j_{n\text{-propanol}}$ on local CO concentration (*Ref 41*: ACS Catal. 2018, 8, 931-937).”

Figure R1-3 (i.e., Supplementary Fig. 28). Plot of the partial current density for n-propanol as a function of the CO partial pressure on Cu adparticle electrocatalysts at -0.36 V vs. RHE, a slope value of 0.9 is obtained.

6) In the same context, proton concentration dependency on n-propanol production also can be a useful key to elucidate the reaction mechanism. While both of *OCCOH and *CCH₂ are currently suggested as the possible source of C₂ precursor, those intermediates necessitates different number of protons for its formation. As the author indicated in the manuscript, since the CO concentration is mainly determined by CO partial pressure in the gas feedstock, this can be done by simply changing the pH of electrolyte.

Author's Reply: We carried out CO-RR tests in various electrolytes with different [KOH], i.e., different proton concentration, pH. With increasing of [KOH] from 0.5M to 1M (corresponding in a pH shift from 13.7 to 14), an increase in C₂₊ products is observed (**Figure R1-4**). These results suggest that increased [KOH] favors the C-C coupling process, consistent with the recent *Science* paper (*Ref 7*: Science 2018, 360, 783-787). However, with a further increase of [KOH] from 1M to 5M (corresponding to pHs of 14 to 14.7), we found that both n-propanol and ethylene productions drop, whereas ethanol generation remains unchanged. Acetate formation also increases sharply with [KOH] in this range. We attribute the acetate increase to the attack of n-propanol/ethylene intermediates in the concentrated OH⁻ solution, in line with Kanan's report in *Nature* (*Ref 13*: Nature 2014, 508, 504-507) and very recent mechanistic report by Goddard and Ager in JACS (J. Am. Chem. Soc. 2018, 140,

9337-9340). Collectively our findings suggest a [KOH] of 1M, with a corresponding pH = 14, to be preferential for n-propanol production.

We now note in the paper (**Page 14, Line 12**):

“A proton concentration dependency on n-propanol production was also examined (**Supplementary Fig. 29**). With a range of 0.5 – 5M KOH, 1M KOH is found to produce the highest $FE_{n\text{-propanol}}$, a further increase of KOH concentration prioritizes the acetate formation due to the attack of n-propanol/ethylene intermediates in the concentrated OH^- solution (*Ref 13*: Nature, 2014, 508, 504).”

Figure R1-4 (i.e., Supplementary Fig. 29). CO-RR product selectivities (FE, %) on Cu adparticle electrocatalysts in different [KOH] electrolytes at ~ -0.47 V vs. RHE.

7) I also would like to point out the possible relationship between acetate and C2 chemicals, and/or acetate and n-propanol. Although the Faradaic efficiency change of acetate is relatively small at all CO partial pressures (Figure 5), however, all of ethylene, ethanol and n-propanol exhibit a general increase of Faradaic efficiency with increasing overpotential, while that of acetate decreases against applied potential (Figure 4b). This may imply subsequent conversion of CO to acetate, then to other C2 and C3 chemicals.

Author’s Reply: The plot of product distribution as a function of applied potential does indeed suggest a possible relationship between acetate and C_{2+} , especially ethylene and n-propanol. As shown in **Fig. 4b**, a quick drop of acetate formation commensurate with the sharp increase of ethylene and n-propanol production is observed at a narrow potential window from -0.3 V to -0.4 V vs. RHE. In contrast, the ethanol generation is only slightly changed. These results suggest the pathways for acetate formation are similar to those for ethylene and n-propanol production, but they diverge at certain points with different applied potentials (energies), consistent with a recent DFT report by Bell and Head-Gordon (*Ref 39*: ACS Catal. 2018, 8, 1490-1499).

Accordingly, we have included this argument in the revised manuscript. **Page 10, Line 14:**

“While both ethylene and ethanol exhibit a general increase of FE with increasing overpotential, the FE of acetate first decreases and then reaches a plateau, which suggests a possible relationship between acetate and C₂₊ chemicals. In particular, a prompt product switchover between acetate and ethylene/n-propanol at a narrow potential window from -0.3 V to -0.4 V vs. RHE, indicates acetate, ethylene and n-propanol may share similar pathways. However, these pathways deviate depending on applied potentials, consistent with previous analysis (*Ref 39: ACS Catal. 2018, 8, 1490-1499*).”

Reviewer #2 (Remarks to the Author):

In this paper, the authors put forward a Cu “adparticle” structure, both computationally and experimentally, as being particularly selective for CO electrocatalytic reduction to n-propanol. They start with a DFT hypothesis that clusters of 2-4 Cu atoms sitting on top of a single crystal Cu surface can potentially serve as active sites for CO reduction and utilize this structure to model the adsorption energies of key intermediates. They then synthesize a Cu “adparticle” catalyst that consists of ~3-5 nm Cu nanoparticles sitting on larger Cu particles and study its CO reduction activity. They compare this system to two other Cu nanostructured catalysts and observe an increase in peak n-PrOH faradaic efficiency from 15% to 23% on Cu AD vs. Cu NB at -0.54 V vs. RHE.

The 23% faradaic efficiency for n-PrOH and partial current density of 11 mA/cm² represents an improvement in absolute selectivity and activity compared to literature examples for this reaction. However, the structural hypothesis that they put forward as being responsible for this change in activity is not strongly substantiated. For modest activity differences like the ones reported here, it is difficult to clearly attribute that difference to a specific structure without a wide range of comparison materials.

Author Reply: We detail below where we have introduced a new suite of experimental studies that seek to address these comments.

A few specific comments are provided below:

1. The DFT calculations put forward a very specific structure, i.e. 2, 3, or 4 Cu atoms sitting on top of a single-crystal Cu surface. While the calculations provide interesting data about adsorption energies, the connection to the experimental samples are quite tenuous. The actual “adparticle” structure studied consists of 3-5 nm nanoparticles on a Cu surface, which represents hundreds of Cu atoms. How does the experimental structure actually relate to the DFT structural model? Is it possible to extrapolate conclusions from a 2-4 adatom structure to an adsorbed nanoparticle with hundreds of atoms?

Author’s Reply: We now clarify that the inclusion of different numbers of adatoms on various Cu basal planes in the simulations was not to address the size effect of adparticles in different cases. Rather, we build a one polycrystalline Cu adparticle model with a possibility of possessing various low-coordinated Cu sites (Supplementary Table 1), to demonstrate the low-coordination effect in general, and the associated enhancement in the C₃ chemical formation, consistent with our experimental results..

In order to simulate bond forming/cleavage of molecules on the surface, we employ here first-principles DFT simulations. Due to computational capacity limitations, it

was not feasible to model larger, 3-5 nm nanoparticles, with hundreds of atoms on the top of a Cu surface. Instead, with a few adatoms on different Cu basal planes, we can include the influence of surface Cu coordination numbers (CNs) on bond forming/cleavage of molecules, and the overall C₃ formation energies during CO-RR in this case.

To make this point clear, we have made several changes to the manuscript:

- (a) **Fig. 1a** has been revised to clearly show the geometry here and, most importantly, the adparticles of interest and their associated coordination numbers. This was not sufficiently clear previously, and the original **Fig. 1** simulation schematics gave the impression that the adparticle size was the key parameter probed, thus highlighting a mis-match with the experiments.
- (b) We are also now clear in the text. **Page 6, Line 4**: “We employed density functional theory (DFT) to assess our hypothesis that Cu adparticles could provide such benefits. We include a different number of adatoms on various Cu basal planes to construct one polycrystalline Cu adparticle model with a possibility of possessing various low-coordinated Cu sites (**Fig. 1a, Supplementary Table 1**).”
- (c) We also note in the text the size discrepancy between the simplified DFT case, and the produced adparticles (**Page 8, Line 21**). “Although the resulting adparticles are significantly larger than those simulated, particles of this size are expected to result in a variety of coordination numbers (*Ref 40*: J. Am. Chem. Soc. 2014, 136, 6978-6986), as exemplified in the simplified, small-particle DFT simulations.”

2. The authors mention that their catalysts have a higher ECSA than oxide-derived Cu samples produced in other ways. To what extent is the ECSA affecting the surface-area normalized current density? Several authors have shown that mass-transport within a high-surface area material can affect the selectivities for H₂ vs. C₁ products vs. C₂ products. Is there also a potential correlation between ECSA and C₂ vs. C₃ products? This may also be consistent with the observed increase in C₃ selectivity with higher CO partial pressures.

Author Reply: We now better explain that Cu AD samples show a higher n-propanol partial current density, both with respect to geometrical current density (**Fig. 4d**) and ECSA-normalized current density (**Supplementary Fig. 26**), than Cu NB and Cu NP species. These differences suggest the increase of intrinsic activity of Cu AD catalysts due to its higher density of under-coordinated sites, i.e., the formation of adparticles. To address the correlation among ECSA, CO partial pressure, and CO-RR product selectivity, we have in-situ derived Cu AD catalysts at different CO partial pressures, measured their ECSAs (**Figure R2-1**), and compared their associated n-propanol selectivities (**Figure R2-2**). The results indicate that Cu AD catalysts derived under various CO partial pressures have similar ECSA values, however, their

n-propanol selectivities exhibit large differences as a function of local CO concentration. We found that in the CO partial pressure range from 15% to 100%, a linear correlation between $j_{n\text{-propanol}}$ and CO partial pressure is identified with a slope value of ~ 1 (Figure R1-3), suggesting the first-order dependence of $j_{n\text{-propanol}}$ on local CO partial pressure.

Figure R2-1. Cyclic voltammograms taken over a range of scan rates and determination of double-layer capacitance on Cu AD samples in-situ derived under 65%

(a, b), 35% (c, d), and 15% (e, f) CO conditions. (g). Measured capacitance values and surface roughness factors of various electrode materials.

Figure R2-2. Plot of n-propanol selectivities (FE, %) on Cu AD electrocatalysts in-situ derived under various CO partial pressures as a function of applied potential in 1 M KOH.

3. The authors hypothesize that increased CO binding at under-coordinated Cu surface sites may be responsible for increased C₃ selectivity on their Cu AD samples. Besides DFT, have the authors been able to measure a CO adsorption energy on their surface?

Author Reply: We pursued CO adsorption energy measurements on the surfaces of different catalysts using the CO-TPD tests reported in literature. Unfortunately, this testing was not conclusive, due to the harsh testing conditions inherent to this technique. We summarize below (also noted in response to Reviewer #1, Question #4):

First, we returned to the lab to prepare enough derived-Cu powder samples for CO-TPD measurements (~100 mg/sample/test). The results of the CO-TPD testing are now shown in **Figure R1-1**. Both Cu AD and NB samples exhibited strong desorption signals at high temperature compared to Cu NP controls; however, it was not possible to confirm that this desorption was CO, and not contamination. Although the samples were stored under inert N₂ gas condition before testing, contamination from carbon species from the gas diffusion layer (porous carbon materials) or partially oxidized copper cannot be ruled out when using this complex integrated sample. We received similar reply regarding the CO-TPD test for this material system from Prof. Ib Chorkendorff at DTU who contributed to the previous CO-TPD work on Cu published in JACS in 2015 (J. Am. Chem. Soc., 2015, 137, 9808).

Second, in order to highlight the significance of adparticle for CO adsorption and hence C₃ formation under CORR experimentally, we conducted a post-annealing process on our Cu AD sample to erase the adparticle texture while maintaining the overall morphology of Cu catalysts (**Figure R1-2**). A shift of product selectivity from

n-propanol to ethylene after annealing is observed compared to Cu AD results in **Fig. 4**, which is due to the thermal removal of the AD texture.

4. The authors compare their samples to a Cu “nanobump” sample and a Cu nanoparticle sample. Since all three of these materials are prepared in different ways and result in very different morphologies, it is difficult to build a structure-activity relationship just from these three materials. Is there any way to systematically vary the ad-nanoparticle size or density using their synthetic method to obtain more systematic structure-activity relationships?

Author Reply: To address this comment and further support the adparticle effect on n-propanol selectivity, we introduced an additional control sample which was prepared by post-annealing of Cu AD samples at 150°C under N₂ gas conditions, denoted Cu AD-150. In this case, Cu AD-150 shares a similar morphology with Cu AD, but the adparticles on Cu AD are erased via thermal annealing (**Figure R1-2**). Consequently, while the total FE_{C2+} on Cu AD-150 remains similar, FE_{n-propanol} drops drastically, to less than 10%, with an increase of FE_{ethylene}.

Accordingly, we added this extra control and associated discussion into our revised manuscript. **Page 11, Line 23:**

“To further exclude additional morphology effects among Cu AD, NB and NP species on CO-RR performance, we thermal-annealed Cu AD catalysts at 150°C under N₂ gas condition. This processing erases the adparticle texture while maintaining the overall morphology (**Supplementary Fig. 27**). A dramatic decrease of FE_{n-propanol} commensurate with the increase of FE_{ethylene} is observed, further supporting the adparticle role in promoting n-propanol production.”

In summary, the authors have put forward a nanostructured Cu catalyst with increased n-PrOH selectivity compared to other nanostructured Cu catalysts. The catalytic data for the increased selectivity is clear, thorough, and compelling. However, the connection between the structure and catalytic activity based on DFT and two comparison samples is much less clear. As a result, I believe this paper would be more suitable for a catalysis specific journal.

Author Reply: We thank Reviewer #2 for prompting us to carry out this important further suite of studies.

Reviewer #3 (Remarks to the Author):

Li et al. present a strategy to enhance the production of 3-carbon products in electrochemical CO reduction on copper by introducing under-coordinated surface copper atoms that enhance the binding of CO and 2-carbon intermediates in close proximity. They motivate this idea using density-functional theory calculations of adsorption and reaction energies, synthesize particles with suitable under-coordinated surface atoms and demonstrate selectivity to 3-carbon products in electrochemical reduction. This manuscript makes an important contribution towards favoring longer-chain products in electro-catalysis, a key scientific challenge, and merits publication in Nature Communications.

Author Reply: We thank Reviewer #3 for this feedback on the work.

Some questions the authors should address prior to publication:

1. Electrochemical environments can strongly affect adsorption and reaction energies, especially by favoring charged surface configurations. This is now increasingly being incorporated in DFT calculations using advanced solvation models and grand-canonical (fixed electron potential) techniques to predict accurate onset potentials (eg. in Refs 9, 28 cited in the manuscript). The authors should justify their choice / acknowledge the limitation of neutral in-vacuo DFT calculations.

Author Reply: We agree with reviewer that in the current work, and address these limitations below, and in the text.

First, regarding solvation, we note from literature examples that include both explicit and implicit solvation (J. Am. Chem. Soc. 2016,138, 483-486; Proc. Natl. Acad. Sci. U. S. A. 2017, 114, 1795-1800) that CO dimerization and H shuttling from H₂O occurs simultaneously and the key C₂ intermediate will be OCCOH during CORR/CO₂RR. We therefore employ here OCCOH as one of the key intermediate for the C₁-C₂ coupling to generate C₃ products during CORR. Also, we compared the reaction energy of $2CO^* \rightarrow OCCO^*$ and $2CO^* + \frac{1}{2}H_2 \rightarrow OCCOH^*$ over Cu surfaces with/without Cu adparticles. The main trend that Cu adparticles over Cu(111), Cu(100), and Cu(211) decrease the reaction energies of CO dimerization to the OCCO species, is as the same as those to the OCCOH intermediate (**Figure R3-1**).

Figure R3-1. Reaction energies of $2\text{CO}^* \rightarrow \text{OCCO}^*$ and $2\text{CO}^* + \frac{1}{2}\text{H}_2 \rightarrow \text{OCCOH}^*$ over Cu surfaces with/without Cu adparticles.

Second, regarding the electron potential, we note that our intention in these simulations is to show the significant effects of Cu adparticles for promoting the C_3 production during CO-RR, rather than predicting accurate onset potentials. We now provide additional simulations including the influence of external fields and surface charging.

Specifically, we added the following to our revised manuscript (**Page 7 , Line 17**):

“In addition, we varied the external fields and surface charge via the Neugebauer and Scheffler method (*Ref 33: Phys. Rev. B 1992, 46, 16067-16080*), and the results support Cu adparticles enhancement of C_3 electroproduction during CO-RR (**Supplementary Fig. 11-13**).”

In addition, we also added the following text in the SI:

“Furthermore, we applied the Neugebauer and Scheffler method to add the applied field and charging surface to examine how the applied field can influence the CO adsorption energies, the reaction energies of CO dimerization and CO-OCCOH coupling over the Cu(111) surface in the presence and absence of Cu adparticles.

As shown in **Supplementary Fig. 11-13**, by increasing the applied field from 0 to 0.8 V/\AA , the adsorption energies of CO molecule on a Cu surface at their most favourable adsorption sites monotonically increased, and the reaction energies of CO dimerization / C_1 - C_2 couplings over a Cu surface monotonically decreased.”

Supplementary Fig. 11. The applied field effects on the adsorption energies of CO over Cu(111) at a fcc site (a), over Cu(111) with two Cu adatoms at a top site (b) over Cu(111) with three Cu adatoms at a top site (c). The adsorption energies of the CO

molecule on the Cu surface at their most favourable adsorption sites monotonically increased as we increased the applied field from 0 to 0.8 V/Å.

Supplementary Fig. 12. The applied field effects on the reaction energy of CO dimerization over Cu(111) (a), over Cu(111) with two Cu adatoms (b) over Cu(111) with three Cu adatoms (c). The reaction energies of CO dimerization on the Cu surface monotonically decreased as we increased the applied field from 0 to 0.8 V/Å.

Supplementary Fig. 13. The applied field effects on the reaction energy of CO-OCCOH coupling over Cu(111) (a), over Cu(111) with two Cu adatoms (b) over Cu(111) with three Cu adatoms (c). The reaction energies of CO-OCCOH on the Cu surface monotonically decreased as we increased the applied field from 0 to 0.8 V/Å.

2. CO adsorbed on metal surfaces is a prototypical example of where DFT can be rather inaccurate, completely mis-predicting the stable adsorption sites and geometries. The authors should acknowledge this limitation as well.

Author Reply: We agree with reviewer that pure PBE (Perdew-Burke-Ernzerhof) functional can be inaccurate in predicting favorable adsorption sites and can over-estimate the adsorption energies of CO. However, RPBE functionals perform well for the adsorption energies of the catalytic intermediates on single-crystal late transition-metal surfaces that involve pure chemisorption (i.e., CO over Cu) (*Ref 48*: J. Phys.: Condens. Matter 2004, 16, 1141-1164; *Ref 49*: J. Phys. Chem. C 2017, 121, 4937–4945). Therefore, in our revised manuscript, we additionally calculate the adsorption of CO as well as CO dimerization over Cu(111) in the presence and absence of Cu adparticles using RPBE functional.

We added the following (Page 16, Line 1):

“We also examined the CO adsorption energy (**Supplementary Fig. 30**) and reaction energy of CO dimerization (**Supplementary Fig. 31**) using RPBE functional, which is more accurate for the catalytic intermediate adsorption on single-crystal late transition-metal surfaces that involve pure chemisorption (i.e., CO over Cu) (*Ref 48*: J. Phys.: Condens. Matter 2004, 16, 1141-1164; *Ref 49*: J. Phys. Chem. C 2017, 121, 4937–4945). The main trend that Cu adparticles

increase the adsorption strength of CO and decrease the reaction energies of CO dimerization is supported by the RPBE functional results.”

We have also added the following paragraph into our **SI**:

“In addition, the RPBE functionals generally perform well for the adsorption energies of the catalytic intermediates on single-crystal late transition-metal surfaces that involve pure chemisorption (i.e., CO over Cu). In this work, we also compared both PBE functional and RPBE functional on the adsorption of CO as well as the reaction energy of the CO dimerization.

As shown in **Supplementary Fig. 30**, the GGA-PBE functional overestimated the adsorption energies of CO over Cu with various Cu adparticles. But a similar trend can be found using both RPBE and PBE functionals, in which the Cu adparticles can significantly increase the CO adsorption energy. As shown in **Supplementary Fig. 31**, the difference in the reaction energies of CO dimerization between PBE functional and RPBE functional is less than 0.1 eV. And the main conclusion that Cu adparticles enhance the CO dimerization is supported by the RPBE functional as well.”

Supplementary Fig. 30. The adsorption energies of CO over Cu(111) in the presence and absence of Cu adparticles using PBE and RPBE functionals.

Supplementary Fig. 31. The reaction energies of CO dimerization over Cu(111) in the presence and absence of Cu adparticles using PBE and RPBE functionals.’

3. The term adparticle seems suitable for the on-average 3 nm sized entities observed in the high-resolution TEM images, while the calculations seem to be on small clusters of adatoms which are less than a nm across. Given this discrepancy in scale, would edges on stepped surfaces be more representative of the active sites in the experimentally-observed structures, rather than islands of few atoms?

Author Reply: We have included the Cu(211) stepped surface and the associated edge atoms in the DFT model (**Supplementary Table 1**). Cu atoms with coordination numbers (CNs) of 7, 9 and 10 are attributed to the step, upper terrace and lower terrace atoms on Cu(211) basal plane. Those edge atoms on stepped surfaces do play meaningful roles to enhance C3 formation as shown in **Fig. 1**. Nevertheless, the variation of Cu CN is limited in this case. The adparticle surface, distinct from Cu basal planes, can allow a broad distribution of Cu CNs (especially on Cu(111) and Cu(100)). The relatively small adparticles simulated here capture the various Cu CNs and Cu basal planes. Further details can be referred to Author’s Reply to Reviewer #1 (Question #1). In addition we note the size discrepancy directly in the text (**Page 8, Line 21**):

“Although the resulting adparticles are significantly larger than those simulated, particles of this size are expected to result in a variety of coordination numbers (*Ref 40: J. Am. Chem. Soc. 2014, 136, 6978-6986*), as exemplified in the simplified, small-particle DFT simulations.”

REVIEWERS' COMMENTS:

Reviewer #1 (Remarks to the Author):

The revised manuscript is improved enough to support the validity and novelty of this work. Moreover, the point-by-point responses are well presented.

Reviewer #2 (Remarks to the Author):

The authors have addressed my concerns.

Reviewer #3 (Remarks to the Author):

The authors have suitably addressed comments from the previous review. In particular, the previous weak point: an unclear connection between the calculations and experiment, has been strengthened substantially in the revised manuscript. It is therefore now suitable for publication in Nature Communications.